# Strategies for Deploying a Sensor Network to Explore Planetary Lava Tubes

**DOI:** 10.3390/s21186203

**Published:** 2021-09-16

**Authors:** Himangshu Kalita, Jekan Thangavelautham

**Affiliations:** Space and Terrestrial Robotic Exploration (SpaceTREx) Laboratory, Aerospace and Mechanical Engineering Department, University of Arizona, Tucson, AZ 85721, USA; hkalita@email.arizona.edu

**Keywords:** sensor network, cave exploration, mapping, navigation

## Abstract

Recently discovered pits on the surface of the Moon and Mars are theorized to be remnants of lava tubes, and their interior may be in pristine condition. Current landers and rovers are unable to access these areas of high interest. However, multiple small, low-cost robots that can utilize unconventional mobility through ballistic hopping can work as a team to explore these environments. In this work, we propose strategies for exploring these newly discovered Lunar and Martian pits with the help of a mother-daughter architecture for exploration. In this architecture, a highly capable rover or lander would tactically deploy several spherical robots (SphereX) that would hop into the rugged pit environments without risking the rover or lander. The SphereX robots would operate autonomously and perform science tasks, such as getting inside the pit entrance, obtaining high-resolution images, and generating 3D maps of the environment. The SphereX robot utilizes the rover or lander’s resources, including the power to recharge and a long-distance communication link to Earth. Multiple SphereX robots would be placed along the theorized caves/lava tube to maintain a direct line-of-sight connection link from the rover/lander to the team of robots inside. This direct line-of-sight connection link can be used for multi-hop communication and wireless power transfer to sustain the exploration mission for longer durations and even lay a foundation for future high-risk missions.

## 1. Introduction

The evidence of a possible entrance to an underlying lava tube was first detected from the 10 m/pixel-resolution Terrain Camera (TC) aboard the Japanese lunar orbiter Selenological and Engineering Explorer (SELENE) in 2009 [1]. The TC data detected three features, namely the Marius Hill Hole (MHH), Mare Tranquilitatis Hole (MTH), and Mare Ingenii Hole (MIH), as shown in Figure 1. The feature Marius Hill Hole (MHH) is a vertical hole in the middle part of a rille (303.3° E, 14.2° N, −1.65 km elevation from the lunar mean radius of 1737.4 km) in the Marius Hills region, a region rich in prominent volcanic features [2]. The hole is 65 m in diameter with an estimated depth of 90 m [1]. The hole is very similar to openings called “skylights” in lava tubes on the Earth and Mars and differs from normal impact craters in the vicinity. The Mare Tranquillitatis Hole (MTH) is located at 33.2° E, 8.3° N, −0.77 km elevation, 350 km from the Apollo 11 landing site. It is nearly twice the diameter of the Marius Hills feature and is roughly circular. The hole was estimated to be over 100 m deep from shadow measurements. The Mare Ingenii Hole (MIH) is located at 166.0° E, 35.6° S, −3.62 km elevation. It has a slightly irregular shape (rounded triangle) with a long-axis length of 140 m (east-west) and a short-axis length of 110 m (north-south) in the TC data.

### 1.1. High-Resolution Lunar Imaging

The high-resolution Narrow-Angle Camera (NAC) images the lunar surface at 0.5 to 1 m/pixel-resolution onboard the United States lunar satellite Lunar Reconnaissance Orbiter (LRO), launched in 2008, providing new details regarding the morphologies of the holes. LRO NAC took very clear, detailed images of MHH and its floor at high solar-elevation angles and confirmed the Marius feature to be nearly circular. From the NAC data, the floor axis lengths were estimated to be 59 m × 50 m with a depth of 47 m [3,4]. The data confirmed that it is neither a normal impact crater nor a volcanic vent as no ejecta was seen surrounding the hole. Moreover, the lowest portion of the inner wall of MHH does not continuously follow the floor, which strongly suggests the existence of extended voids (possibly lava-tube caverns) at the bottom of the hole. Studying the geometry of the shadow on the floor suggests that the floor is flat as it is quite similar to the outline of the inner edge of the hole. Although the areas surrounding MHH are 3.56 Gyrs old [5], at least one layer under the surface formed 0.1 Gyrs earlier than the surface [1]. This sheds light on the possibility of a lava tube formed from the lower flow, around 3.6 Gyrs ago, corresponding to a time of high lunar volcanic activity [6]. Although the age of formation of lava tubes has been estimated, the age of formation of the hole structure is still unclear. However, it could be inferred that the hole structure formed possibly because the roof collapsed. The pile of rubble from the collapsed roof at the opening of the hole might have been crushed and flattened by meteorite bombardment and covered by regolith and soil from the inner walls. The meter-scale boulders seen from the NAC data on the floor may have fallen from the inner wall in recent times.

The LRO NAC images of the Mare Tranquillitatis Hole and Mare Ingenii Hole also clarified the details of these features. The long axes of MTH and MIH are measured to be 98 m and 118 m, respectively, and the short axes to be 84 m and 68 m. The depths are measured to be 107 m for MTH and 45 m for MIH from shadow measurements based on the LRO NAC images [3,4]. No ejecta or boulders are seen around both the holes, implying that the holes are not volcanic vents. The surface surrounding MTH was formed 3.62 Gyrs ago, while two formation ages of the surrounding area of MIH are deduced, 3.17 Gyrs and 3.59 Gyrs, corresponding to a time of large amounts of lunar mare volcanism. As with MHH, the inner walls of both MTH and MIH do not follow the floor, which strongly suggests the existence of subsurface voids. Additionally, more than 200 pits were also discovered in the impact melt deposits.

### 1.2. Potential Utility of Lunar Lava Tubes

These holes have high potential as resources for scientific study as various important geological and mineralogical processes could be uniquely and effectively observed inside these holes. Although exposed lava layers could be observed on the inner walls of craters and rilles, they are covered with regolith from the surface. However, since the inner walls of these holes are nearly vertical, they exhibit fresh surfaces with a much larger section, thereby preserving more lunar volcanic history. Moreover, there is a possibility of water storage inside these lunar holes. Hydroxyl and/or water molecules can be produced by reactions of oxygen-bearing lunar materials and solar-wind protons [7]. A very speculative estimate is made for the maximum column density of the protons inside MHH, MTH, and MIH to be 0.24, 0.24, and 0.19 tons/m^2^, possibly corresponding to water molecule densities of nearly 2.1, 2.2, and 1.8 tons/m^2^ [8].

In addition to scientific interest in geological and mineralogical studies, lunar holes are excellent candidates for the placement of future lunar bases. For both unmanned and manned activities on the surface of the Moon, one of the most serious concerns is radiation damage due to solar and galactic cosmic rays (SCR and GCR). On the lunar surface, the doses of radiation exceed the allowance limit of 1.2 Sv per lifelong for JAXA astronauts (“JAXA radiation control guideline for astronauts of the International Space Station” 2002), and far in excess of the annual dose recommended by the United States Nuclear Regulatory Committee (1 mSv/yr) (“U.S. Nuclear Regulatory Commission Regulations: Title 10, Code of Federal Regulations, Subpart D” 1991). Due to the limited field of view (FOV) from the bottom of these holes, they can definitely reduce the effects of cosmic rays. For a hole of the same scale of depth and diameter as MHH and MTH, the amount of GCR, including protons and heavy ions entering the hole, is less than about 10% of that striking the surface. The total dose rate of p-GCR at the floor would be less than 10 mSv/yr. The rate of n-GCR induced by p-GCR (protons and heavier ions) and heavy ions would be a maximum of several tens of millisieverts per year during the worst days at solar minimum [8]. In addition to irradiation by cosmic rays, meteorite impacts and temperature fluctuations pose a threat to astronauts and instruments operating on the lunar surface. Lunar holes would provide good protection from space particles, with the damage probability at the bottom of the hole an order of magnitude less than that on the surface. The holes will also guard astronauts and facilities from the danger of hyper-velocity secondary impacts of ejecta that are splattered by primary impact events. The lunar surface temperature varies widely, for instance, 102 K to 384 K at the Apollo 15 landing site in the Hadley rille (26° N, 3.6° E) [9] and 92 K to 374 K at the Apollo 17 landing site in the Taurus Littrow valley (20° N, 30° E) [10]. However, the temperature in a shadowed portion of the hole bottom ranges from 253 K to 303 K (−20 °C to 30 °C) in one lunar day, but the temperature ranges from −170 °C to 110 °C at the surface [8]. If the floors permit mobility, one can secure even smaller doses of radiation, lower probability of damage from meteorite impacts, and smaller temperature variations.

### 1.3. Martian Lava Tubes

Lava tubes and related flow features on the surface of Mars were first recognized by analyzing images from the Viking orbiter. More recently, lava tube entrances have been discovered using the Thermal Emission Imaging System (THEMIS) onboard the 2001 Mars Odyssey orbiter. Other discoveries have been made by the Mars Orbiter Camera (MOC) onboard the Mars Global Surveyor, the High-Resolution Stereo Camera (HRSC) onboard the Mars Express orbiter, and the High-Resolution Imaging Science Experiment (HiRISE) onboard the Mars Reconnaissance Orbiter (Figure 2). On Pavonis and Ascraeus Montes, lava tubes appear to have collapsed, leaving linear, sometimes discontinuous, channel-like features. Several lava tubes have been identified on the northern portion of the Northern shield Alba Patera between 241–2451 W and 44–511 N [11]. In the Tartarus Colles, a partially roofed lava channel was imaged by the HiRISE camera, indicating this feature was a lava tube with a hollow conduit-like channel [12]. Several lava tubes and flow channels were identified on Olympus Mons based on MOC images [13]. Tube-fed flows and raised ridges (interpreted as uncollapsed lava tubes) cover as much as 8% of the flanks of Olympus Mons based on HRSC images [14]. Caves have also been identified mainly by the presence of “skylights”, which can originate as openings in partially collapsed ceilings of lava tubes [15]. If life exists on Mars, one compelling place to search for it would be within a lava tube or cave [16,17]. Here, living organisms could have been sheltered from dangerous ultraviolet radiation, while volcanic minerals might have provided a rich source of nutrients for chemosynthesis (similar to living colonies near Earth-based volcanic vents) [18]. Even more intriguing is the possibility the caves were partially or wholly sealed, permitting the build-up of a high-pressure atmosphere and the existence of liquid water. Minerals commonly formed in Earth caves may also be found in Martian caves and may represent important records of past environmental conditions or even past life through preservation of biosignatures [19].

### 1.4. Robotics for Exploring Rugged Environments

The rugged terrain inside these voids is due to the presence of detritus from the collapsed roof materials and steep-walled vertical entrances, making it impossible for conventional wheeled robots to explore them. Therefore, there is an important need for the development of next-generation robotic and autonomous systems that can explore these extreme and rugged environments, which has been prioritized by NASA as outlined in the 2015 NASA Technology Roadmaps [20]. There have been several compelling strategies proposed to explore these pit and lava tubes. These include a wheeled secondary robot connected to a tether that is deployed from a larger rover called Axel [21]. Axel would roll down a cliff or pit into a rugged environment. Another variation called the DuAxel consists of a pair of Axels robots that operate as a four-wheel rover [22]. In addition, the DuAxel can transform by having each of the Axel robots stretch along a tether, with one of the Axels anchored and a second Axel rolling down a cliff. In both situations, concern arise about the suitability of long tethers for climbing or rolling down cliffs. Martian and lunar surface contain rocks that are sharp and rough with properties of crushed glass. These surfaces could breach or even cut into typical tether materials. A tether reinforced with hardy material also poses concerns. These tethers could scrape and damage the entrance pit walls that are hundreds of millions if not billions of years old. Alternative configurations have been analyzed, including lowering one half of the DuAxel platform from a tether platform anchored across the top of the pit [23]. This new configuration could prevent the DuAxel platform scraping and damaging the pit walls. What is also required for this configuration is two anchoring points across the top of the pit. To anchor a tether that will support raising or lowering a rover down into the pit requires substantial strength, and hence the anchoring systems need to work well in this relatively unknown lunar or Martian environment.

Our approach is to develop an architecture to explore these offworld lava-tube environments with the use of Commercially-Off-The-Shelf (COTS) technology. The rise of CubeSats for low-Earth orbit missions, technology demonstration missions, and even interplanetary missions has led to the rapid advancement in electronics, sensors, actuators, instruments, and power supplies [24]. Further technological advancement leads to radiation-hardened versions of these components for use in deep-space and planetary environments. The architecture is presented as a spherical robot called SphereX that is designed to perform ballistic hops for mobility, as shown in Figure 3. Each robot is 4 kg in mass and has a diameter of 30 cm. Multiple SphereX robots can be carried by a large rover or a lander and can be strategically deployed near scientifically important environments which are inaccessible to the mother rover or lander [25,26,27]. Unlike the Axle and DuAxle platforms, SphereX will not require a hardy anchor system that needs to be tested and validated in a relatively unknown surface environment to work.

SphereX contains electronics, sensors, and actuators equivalent to CubeSat components. The electronics, sensor, and actuator package consist of a computer for command and data handling, a radio and multiple antennas for communication, a power management board and battery pack for power, an array of guidance, navigation and control sensors and actuators, and a volume for 1 kg science payload. For mobility, SphereX can perform unconventional mobility in the form of ballistic hopping and rolling with the help of a hybrid mobility system comprising of a single-thruster miniaturized propulsion system and an Attitude Determination and Control (ADCS) System. Due to reduced traction on low-gravity environments, hopping mobility has an advantage over wheeled and rolling mobility as the surface interaction time is minimized. Moreover, a robot performing hopping mobility can overcome large obstacles much higher than the robot itself compared to wheeled and rolling mobility.

Comparatively speaking, hopping mobility has advantages over wheeled/rolling mobility but at the expense of higher energy consumption. However, it has a definite advantage while exploring rugged environments as robots with wheeled/rolling mobility may very easily get stuck. Navigation is performed using the onboard 3D LiDAR, which maps the environment and autonomously generates trajectories. Once a SphereX robot is on the ground, the onboard cameras would take precise, high-resolution panoramic and stereo panoramic images. Furthermore, these images may be used for mission science and public outreach. Stereo video generated during flight can be post-processed to be viewable using VR (Virtual Reality) headsets, giving the public and mission planners the impression of being on site.

The feasibility of successfully exploring a cave or a lava tube depends on an architecture where multiple SphereX robots work collaboratively for navigation and mapping, performing science experiments, and communicating data to and back from a base station. Due to the physical structure of caves and lava tubes, maintaining a direct line-of-sight link from a base station outside to robots inside the caves is nearly impossible. As a result, the robots need to strategically place themselves much like a bucket-brigade and pass information from one to their neighbor and eventually to the base station. A robot team forming a communications bucket-brigade can also be used to transmit power wirelessly using lasers from the base station to the robots inside the cave.

## 2. Mission Architecture

Large-wheeled rovers such as NASA’s Mars Science Laboratory (MSL) and the Mars 2020 rovers (Perseverance) are mechanically limited in the kinds of terrain they can traverse. They can climb obstacles only a few times their wheel diameter, and the wheels can slip in lower-gravity environments [28,29]. Lateral terrain sloping can cause rollovers and must be avoided [30]. Similarly, the maximum climb angle of a rover is limited by gravity and regolith conditions [31]. Areas with hazardous terrain may be ignored completely to reduce mission risk. The fact that so many valuable science instruments are packed into a single platform means MSL will not attempt science unless it is low risk. This approach has the benefit that only one power and mobility system is required for multiple science experiments, but the drawback that risky yet rewarding experiments will never be attempted. Driving a multibillion-dollar rover into a cave and hoping it could autonomously explore and exit the cave is not even considered.

Multiple SphereX robots could be loaded onto a rover. A rover could carry these spheres to hazardous yet scientifically bountiful areas (caves, lava tubes) and use the SphereX to collect science data. These spheres would overcome many of the challenges present for rovers. Hopping would allow climbing over very large obstacles. With the robot being spherical, rollovers are not a problem. Steep slopes can be avoided entirely by hopping over them to level areas. The rover would have hemispherical sockets for the spheres to sit in.

A second example is a mission concept called Arne [32,33] that would consist of a lunar lander containing three SphereX robots that will explore a pit on the Moon, as shown in Figure 4. The lander would descend nearby a lunar pit and deploy the robots one by one using a spring deployment system. The SphereX robots will then hop near the pit entrance, enter the pit and start mapping using the onboard 3D LiDAR sensors and stereo cameras. The robots would act as a communications relay by forming bucket brigades [34] and allowing exploration of areas with vertical overhangs, such as pits and lava tubes. The lander will serve as the main communications relay to Earth and would have instruments such as a science camera to observe the descent and the vicinity around the landing.

Under such a lander/rover-based architecture, the SphereX robots would be given an objective and would work towards it autonomously (baseline design) while streaming data back to ground control. The 15+ min of latency to communicate with Mars means autonomy is required. In real-time, the SphereX would assist a lander or rover to cooperatively build a 3D map of the objective area for later analysis. On a mission to the Moon, operators may opt to control SphereX themselves or let them autonomously explore and observe the streams. Upon seeing an interesting formation in the stream, a scientist would be able to take control of the sphere and perform scientific analysis. After collecting sufficient science data or when power reserves run low, the SphereX robots will return to the rover to be recharged. Another possibility is to stream power inside the pit/lava tube using lasers that the SphereX robots can use to recharge themselves.

### Preliminaries

To understand the feasibility of our proposed concept of using multiple SphereX robots assisted by a rover/lander, we first developed a 3D model of a prospective cave/lava tube with a vertical entrance using Blender software, as shown in Figure 5a. These models have been developed utilizing terrestrial lava tubes and have been scaled up in size based on pit photos to the expected size on the Moon. Although lava tubes differ in size and cross-section, the low gravity of the Moon and Mars is hypothesized to have created larger lava tube cross-sections. These scaled-up lava tube models will then be used for testing our mapping, navigation, and sensor placement algorithms. For sensor placement, multiple sensors (SphereX robots) will be placed along the cave such that they can maintain a direct-line-of-sight connection link, as shown in Figure 5b. This link will then be used for communication and power transfer within the robots.

## 3. Mapping and Localization

For Mapping and Localization, we use an Iterative Closest Point (ICP)-based Pose-Graph Simultaneous Localization and Mapping (SLAM) algorithm on scans generated by the onboard LIDAR sensor [35]. The algorithm exploits the scan registration algorithm and graphical model optimization to build an ICP-based localization system in an unknown environment for SphereX equipped with LIDAR sensors, as shown in Figure 6. It consists of two main layers: an ICP layer and a Graph layer. The keyframes $K_{i}$ and their associated scans form the ICP layer. In the graph layer, nodes $n_{i}$ are associated with each keyframe. The first keyframe $K_{0}$ is associated with the first acquired scan and the associated node $n_{0}$ is created in the graph. Moreover, the initial robot pose with respect to the world origin is added to the pose graph. While the robot moves to acquire local maps, the robot frame R is always expressed with respect to the closest keyframe. At every instant a new scan is acquired, the ICP layer is initiated, which runs the ICP algorithm to find a transformation to align the new scan with the current local map. A threshold value is defined, which decides if a new keyframe associated with the new scan is to be created or the scan to be discarded. If the overlap is lower than the threshold, the scan is accepted; otherwise, it is rejected. If the scan is accepted and a new keyframe is created, the robot pose is corrected, and the new frame and the transformation between the new and the former keyframes are added to the graph as a node. Finally, the local map is rebuilt by merging the newest keyframe with the current local map. The system always searches for potential loop closure, and when it detects one, a local map is built between the loop closing keyframes using the ICP algorithm. If the overlap between them is below the threshold, the new transformation between the loop closing keyframes is added to the graph. This triggers an optimization sequence that repositions the keyframes using the optimization results. Finally, the local maps are reconstructed using the new repositioned keyframes, and the robot pose in the current keyframe is also updated.

### 3.1. Pit Survey Phase

Once the SphereX bots arrive near the pit entrance, a survey of the pit entrance is conducted by hopping from one side of the pit to the other side. Multiple hops will be performed to map the pit entrance by using the ICP-based Pose-Graph SLAM algorithm, and the maps generated during each hop are registered together to estimate the depth and map of the pit floor. These hopping points will be manually selected and communicated to the robots from a ground station.

Figure 7a shows the trajectories of SphereX for surveying the pit with the hopping points for four preselected hops. Figure 7b–e show the maps generated of the pit entrance during each hop. Figure 7f shows the final map of the pit entrance by registering each map generated during each hop. It can be seen that a fraction of the pit floor is mapped, which will then be used for planning the pit entrance phase.

### 3.2. Pit Entrance Phase

Once the floor of the pit entrance is mapped, a suitable point is selected for the robots to land inside the pit from the surface. The suitable point will also be selected manually once the map generated during the pit survey phase is transmitted to a ground station. The robot then hops from the pit surface to the selected point on the pit floor using the soft-landing ballistic hopping mode. As the robot travels, a map of the pit entrance is created while estimating the pose (position and orientation) of the robot using the ICP-based Pose-Graph SLAM algorithm.

Figure 8a shows the initial position and selected final position, along with the trajectory of the robot. Figure 8b shows the map generated during the pit entrance phase along with the estimated trajectory of the robot. Figure 8c,d show the actual and estimated position along the x, y, and z-axis, and the quaternion parameters of the robot, respectively. It can be seen that the position and orientation of the robot were successfully estimated using the ICP-based Pose-graph SLAM algorithm.

### 3.3. Multi-Robot Mapping

With multiple robots deployed inside the cave, a framework is developed to maintain a master map by merging maps generated by each robot [36]. Each robot $R^{i}$ generates its local map $m_{i}$ and pose estimates $p_{i}$ from its 3D Lidar scans. The local maps and pose estimates generated by each robot are then sent to a master robot that generates the master map $M_{t}$ by performing pose-graph optimization. Once the pose-graph is optimized by the master, the trajectory updates $u_{i}$ are transferred back to each robot which is then used to correct their local maps and pose estimates, as shown in Figure 9. We experimented with three robots deployed inside the cave model, as shown in Figure 10. Figure 10a–c show the individual maps generated by each robot 1, 2, and 3. Figure 10d shows the master map generated by using the multi-robot mapping framework.

## 4. Line-of-Sight Analysis

For the architecture of deploying multiple SphereX robots assisted by a large rover or a lander to work, a direct line-of-sight connection link from the base robot on the surface of the pit to the farthest robot inside the pit should be maintained. This section provides an analysis of the feasibility of maintaining such a connection link. The first sub-section presents an analysis of the feasibility of maintaining a connection link from a SphereX robot on the surface of the pit with another SphereX robot on the floor of the pit. The next sub-section provides a framework to find the optimal positions of multiple robots inside the cave/lava tube extension of the pit entrance to maintain such a connection link. The following sub-section then extends the framework for an unknown cave by implementing an Explore → Place Sensor cycle.

### 4.1. Line-of-Sight from Pit Surface to Pit Floor

With the communication and power transfer architecture defined in Figure 5b, the SphereX robot on the surface of the pit needs to maintain a direct line-of-sight connection link with the SphereX robot on the floor of the pit entrance. To maintain a direct line-of-sight connection, link the SphereX robot on the surface near the pit entrance at a distance a deploys a boom of height h vertically upwards that consists of an antenna at its tip. The antenna has a half-power beamwidth B and is oriented at an angle ϕ with respect to the boom, as shown in Figure 11. The distance d on the floor of the pit within which a direct line-of-sight communication link is possible with the SphereX on the surface is calculated by projecting the antenna’s beamwidth on the floor. The dimensions of the pit entrance are approximately equal to that of MTH, as shown in Figure 11.

Figure 12 shows the variation of distance d on the floor of the pit within which a direct line-of-sight communication link is possible with respect to the deployed boom height h in meters and orientation of the antenna ϕ in degrees. The parameters used for the results are Figure 12a: Beamwidth B=30° and a=5 m, Figure 12b: Beamwidth B=60° and a=5 m, Figure 12c: Beamwidth B=30° and a=10 m, and Figure 12d: Beamwidth B=60° and a=10 m. It can be seen that when the surface SphereX is deployed within 5 m of the entrance, a boom of 3 m will be sufficient to maintain a direct line-of-sight communication link with another SphereX on the floor of the pit. In comparison, when it is deployed at a distance of 10 m, the minimum required boom height increases to 5.7 m.

Figure 13 shows the feasibility of a direct line-of-sight communication link with respect to the deployed boom height h in meters and orientation of the antenna ϕ in degrees for an optical transmitter where the angle *B*→0. The parameters used for the results are Figure 13a: a=5 m and Figure 13b: a=10 m. It can be seen that when the surface SphereX is deployed within 5 m of the entrance, a boom of 3 m will be sufficient to maintain a direct line-of-sight communication link with another SphereX on the floor of the pit, while when it is deployed at a distance of 10 m, the minimum required boom height increases to 5.7 m.

### 4.2. Optimal Sensor Placement for Direct Line-of-Sight

We consider a multi-hop relay channel for communication and power transfer inside a cave, comprising of a source device S, a destination device D, and N relays in between the source and the destination device such that a direct line-of-sight link is possible between the source device S and destination device D. Let us consider that the cave is represented by a non-convex polyhedron ℙcave which is the configuration space. Let each obstacle inside the cave be represented by multiple non-convex polyhedrons ℙobs(i);i=1,…,n, n being the number of obstacles. Thus, the obstacle region is denoted by ℙobs=ℙobs(1)∪ ℙobs(2)∪ …∪ ℙobs(n). The obstacle-free space is then denoted by ℙfree=ℙcave\ℙobs. The position of the source device is represented by xinit and that of the destination device by xgoal, which are both an element of ℙfree. The problem is to find the position of the N relays such that the number of relays N is minimized while each sensor has a range *r*. Finding the minimum number of relays N is equivalent to finding the position of each relay such that the total distance between the source S and the destination D through N hops is minimum considering the path between two successive sensors is a feasible path, i.e., the path between them lies in the obstacle-free space ℙfree. We used the concept of Rapidly Exploring Random Tree (RRT) to find the position of the N relay sensors [37]. Although RRT is a common option that creates a graph and finds a direct line of sight path between the source device *S* and destination device D, the path will not necessarily be optimal. However, a modified version of RRT called RRT* aims to provide an optimized shortest path, whether by distance or some other metric that will be used here [38]. The algorithm is initialized with a graph that includes the position of the source device, xinit and no edges. At each iteration *i* = 1,…,*k*, a point xrand∈ℙfree is sampled. An attempt is made to connect the nearest vertex v∈V in the tree to the new sample. If such a connection is successful, xrand is added to the vertex set, and (v,xrand) is added to the edge set. It also considers connections from the new vertex xnew to vertices in Xnear, i.e., other vertices that are within the range r of each sensor. However, not all feasible connections result in new edges being inserted into the edge set E. In particular, (i) an edge is created from the vertex in Xnear that can be connected to xnew along a path with minimum cost, and (ii) new edges are created from xnew to vertices in Xnear, if the path through xnew has lower cost than the path through the current parent; in this case, the edge linking the vertex to its current parent is deleted to maintain the tree structure.

Figure 14 shows the optimal position of sensors inside three different cave models with multiple obstacles (shown by red polygons) inside. The tree maintained is shown in green at 500 iterations, black squares show the sensor positions, and the black line connecting them is the direct line-of-sight connection links. It can be seen that the RRT* algorithm was able to find a feasible connection link from the source device to the destination device.

### 4.3. Sensor Placement in an Unknown Environment

For sensor placement inside a long cave where the entire map of the cave is unknown, the robots will employ an Explore→Place Sensor cycle. Considering the unknown map of the entire cave as ℙcave, the robots explore a subset of the cave ℙcave(1), solves the above algorithm to find the optimal placement of the minimum number of sensors to maintain a direct line of sight connection throughout ℙcave(1) and then moves forward to explore another subset ℙcave(2). Figure 15 shows the placement of sensors inside a cave over five successive Explore → Place Sensor cycles. It can be seen that it was able to find a continuous line-of-sight connection link from the start to the end of the cave.

## 5. Communication

With multiple sensors laid down along the cave, a direct line-of-sight connection link is maintained from a robot at the surface of the pit to the farthest robot inside the cave; these robots can then be used to relay data out of the cave and vice versa. This section provides an analysis of the performance of such a multi-hop communication channel. The data rate, *B_r_* between a transmitter and a receiver for an Additive with Gaussian Noise (AWGN) channel is bounded by Equation (1) [39].
(1)$$B_{r}=W\log\left[1+\frac{P_{t}G_{t}G_{r}}{N_{0}d^{2}}\right]$$
where W is the channel bandwidth in Hz, $P_{t}$ is the transmit power, $N_{0}$ is the noise floor level, $G_{t}$ and $G_{r}$ are the transmitter and receiver antenna gain, and d is the distance between the transmitter and the receiver. Moreover, the bit error rate (BER) describes the number of bit errors per unit time. For an AWGN channel with quadrature phase-shift keying (QPSK) modulation, the BER is a function of the normalized carrier-to-noise ratio $E_{b}/N_{0}$ (energy per bit to noise power spectral density ratio) and is defined as Equation (2) [39].
(2)$$BER=\frac{1}{2}erfc\left(\sqrt{\frac{E_{b}}{N_{0}}}\right)$$
where erfc is the error function, multiple sensors are placed between the first transmitter and the last transmitter for a multi-hop communication link inside a cave. With the transmit power amplified at each sensor, the noise floor level $N_{0}$ also gets amplified, and hence the data rate for the *n^th^* sensor is defined in Equation (3).
(3)$$B_{r}=W\log\left[1+\frac{P_{t}G_{t}G_{r}}{nN_{0}d^{2}}\right]$$

The BER for the multi-hop communication link over N hops and each channel assumed to be independent is defined as Equation (4).
(4)$$BER_{multi-hop}=1-\left(\left(1-BER_{1}\right)\left(1-BER_{2}\right)\ldots \left(1-BER_{N}\right)\right)$$

Assuming each BER << 1, the above expression is approximated as Equation (5).
(5)$$BER_{multi-hop}=\overset{N}{\underset{i=1}{\sum }}BER_{i}$$

However, with the noise amplified, the BER for the *n^th^* sensor is defined as Equation (6).
(6)$$BER_{n}=\frac{1}{2}erfc\left(\sqrt{\frac{E_{b}}{nN_{0}}}\right)$$

The BER for the multi-hop channel is then calculated as Equation (5). For our simulations, we considered three frequencies (a) 500 MHz RF (radio frequency), (b) 2 GHz RF, and (c) 200 THz optical. The 500 MHz and 2 GHz channels were simulated for both 10% and 25% bandwidth, and the 200 THz channel for 1% bandwidth. Figure 16a shows the variation of data rate over a distance of 1000 m for each combination of frequency and bandwidth when the sensors are placed at a distance of 100 m. It can be seen that a data rate of 18 kbps is possible for 500 MHz, 10% bandwidth RF communication channel, while the data rate increases to 0.7 Gbps for 200 THz, 1% bandwidth optical communication channel. Although the data rate does not vary much as the distance between the sensors varies in the orders of 10 s of meters, the BER has a significant impact. Figure 16b shows the variation of BER over $E_{b}/N_{0}$ in dB for different numbers of hops over a distance of 1000 m.

## 6. Power Transfer

The robots laid down along the cave can also be used for wireless power transfer by beaming a high-power laser from the lander/rover to the robots inside the cave through a multi-hop channel. Wireless power transfer through laser is modeled in three phases: (a) Electricity to laser conversion where electrical power input is converted into a laser beam; (b) Laser transmission where laser power is attenuated while traveling through a medium; and (c) Laser to electricity conversion where the laser beam is converted into electrical power by photovoltaic panels.

### 6.1. Electricity to Laser Conversion

Electrical power $P_{s}$ is provided by a power supplier to the laser transmitter, which depends on the current $I_{t}$ and voltage $V_{t}$ such that *P_s_ = I_t_V_t_*. The supply power stimulates the gain medium to generate a laser beam of power $P_{l}$ which relies on current $I_{t}$ as shown in Equation (7) [40].
(7)$$P_{l}=\zeta \frac{h\nu }{q}\left(I_{t}-I_{th}\right)$$
where ζ is the modified coefficient, h is the Plank’s constant, ν is the laser frequency, q is the elementary charge constant, and *I_th_* is the current threshold. Thus, the electricity to laser conversion efficiency is given as Equation (8).
(8)$$\eta_{E-L}=\frac{P_{l}}{P_{s}}$$

### 6.2. Laser Transmission

Laser power is attenuated when transmitted over large distances while propagating through a medium, and the attenuation coefficient *α* is modeled as Equation (9) [40].
(9)$$\alpha =\frac{\sigma }{\kappa }{\left(\frac{\lambda }{\chi }\right)}^{-\rho }$$
where, λ is the wavelength of the laser, *σ* and χ are two constants, *κ* is the visibility, ρ is the size distribution of the scattering particles which depend on the visibility. The laser transmission efficiency is then defined as Equation (10).
(10)$$\eta_{T}=e^{-\alpha d}$$
where d is the distance.

### 6.3. Laser to Electricity Conversion

Under laser illumination of incident power $P_{i}$, the short circuit current $I_{sc}$ of a photovoltaic cell is given by Equation (11) [40].
(11)$$I_{sc}=\frac{qP_{i}}{h\nu }QE$$
where QE is the external quantum efficiency at the wavelength of interest, and for higher efficiency solar cells, the quantum efficiency is very close to unity. The I-V curve model of solar cells is given by Equation (12).
(12)$$I=I_{sc}-I_{sat}\left[exp\left(\frac{V+R_{s}I}{V_{T}}\right)-1\right]-\frac{V+R_{s}I}{R_{sh}}$$
where I is the operating current, *V* is the operating voltage, *V_T_ = nkT/q*, n is the diode factor, k is the Boltzmann constant, *T* is the temperature, *R_s_* is the series resistance, and $R_{sh}$ is the shunt resistance of the solar cell (*R_s_* is assumed to be small and neglected). Thus, the laser to electricity conversion efficiency is defined as Equation (13).
(13)$$\eta_{L-E}=\frac{IV}{P_{i}}$$

### 6.4. Power Transmission over a Multi-Hop Channel

For power transfer inside a cave where the shape of the cave and obstacles makes it impossible to maintain a direct-line-of-sight connection over distance, a multi-hop channel is proposed where the laser beam is relayed from one point to another point through multiple sensors in between. The laser power transmitted by the source, $P_{l\left(0\right)}$ for supplied power, $P_{s}$ is expressed as Equation (14).
*P_l(0)_ = **η_E−L_P_s_*(14)

The incident laser power received by the *n*th sensor is expressed as Equation (15).
(15)$$P_{i\left(n\right)}=\;\eta_{T}P_{l\left(n-1\right)}$$

The electrical power produced by the $n^{th}$ sensor is then expressed as Equation (16).
(16)$$P_{e\left(n\right)}=\eta_{L-E}P_{i\left(n\right)}$$

Each sensor receives the incident laser power from its previous neighbor, then it converts a fraction of it for storage and reflects the rest to the next neighboring sensor. If the efficiency of reflection is defined as $\eta_{R}$, the laser power transmitted by the *n*th sensor to the $\left(n+1\right){\mathrm{th}}^{}$ sensor is expressed as Equation (17).
(17)$$P_{l\left(n\right)}=\eta_{R}\left(P_{e\left(n\right)}-\frac{P_{sensor}}{\eta_{L-E}}\right)$$

Figure 17 shows the variation of transmitted power by each sensor placed at a distance of *d* = [50, 100, 250] m, over a total distance of 1000 m for lasers with wavelength 810 nm and 1550 nm, respectively. For each distance d and wavelength, the model was simulated for three values of initial power supply *P_s_* = [100, 250, 500] W with each sensor storing power at $P_{sensor}=10$ W. Table 1 shows the description of the legend for Figure 17.

## 7. Discussion

In this paper, we outlined strategies for exploring Lunar and Martian caves using an embodiment of a mother-daughter architecture for landers and rovers assisted by multiple SphereX robots. This new embodiment enables a rover or lander to tackle more complex missions and access extreme environments such as caves and lava tube extensions of pit entrances on the surface of the Moon and Mars that would otherwise not be possible. The potential science outcomes can be significant. We have shown that a daughter-craft such as the SphereX platform can perform ballistic hopping that would otherwise not be feasible with a large lander or rover. Notably, the mother-daughter architecture compartmentalizes both risks and capabilities. The mother-daughter architecture is also modular, where one or more daughters (SphereX) can be stored or packaged into freely available space on a lander, rover, or disposal landing component. With each SphereX robot equipped with 3D LiDAR sensors, it is possible to conduct overhead surveying and mapping of the pit entrances, which can then be used to plan the entire mission. Maps generated from overhead surveying can be used to select appropriate landing locations on the pit floor, which will allow the extent of any cave/lava tube extensions if present. If a cave/lava tube extension is present multiple SphereX robots can then be deployed such that they form a multi-hop direct line-of-sight connection link which can then be used for both communication and wireless power transfer.

The SphereX platforms, as outlined, can form communication and power transfer bucket brigades to extend the range and zone of accessibility. This would be advantageous for a lander that would otherwise be static and able to sense and map the surrounding of an area of 500 m^2^ or less. For a rover, the platform can enhance the capability of the mission by traversing hundreds of meters inside a cave/lava tube extension which will otherwise not be possible. The SphereX platforms can also operate in parallel or cooperatively with the landers or rovers. In parallel, the SphereX and lander/rover would perform different science observations and experiments, but in a shorter time, reducing mission operations costs. The SphereX robots can also be modified to make them suitable for sample collection as part of a sample caching mission. The robots can disperse and scout larger areas in a shorter time than would be possible with current rovers. Samples could be identified and fetched back to the rover/lander. The rover/lander would need to pick up the sample package using its prospective robot arm and store it in its caching chamber.

Thanks to mother-daughter architecture, the outlook and mission possibilities are multiplied. Importantly, the risks are compartmentalized thanks to these SphereX robot platforms that would be low-cost and disposable. Effective use of the daughter-craft could multiply the accessible range and the science’s quality and effectiveness for the entire mission. Such an architecture enables access to vast tracts of the Moon, Mars, and other bodies that are presently not accessible. Importantly, it allows large, flagship missions to pursue high-risk, high-reward science without compromising an entire mission.

## 8. Conclusions

In this work, we have proposed strategies using a new mother-daughter architecture for rovers and landers to explore extreme and rugged environments off-world, such as cave and lava tube extensions on the surface of the Moon and Mars. Exploring these harsh, rugged environments will likely provide the best evidence yet of the early formation of the planetary bodies and moons, together with vital clues of how these bodies evolved to their present state. Moreover, exploring these sites could ascertain the range of conditions that can support life and identify planetary processes responsible for generating and sustaining habitable worlds. Unfortunately, current landers and rovers cannot access these sites as they are large, contain many sophisticated instruments, and require hopping or flying to these locations. Instead, we propose using SphereX robot drones to hop into these rugged environments without risking a high-value rover or lander. The SphereX robots would operate through a mix of autonomy and human interaction and perform science tasks complementary to the large rover or lander, such as getting closer to a site of interest and obtaining high-resolution images, 3D maps, spectroscopy, and samples.

These small robots would also perform complementary operational tasks. This includes overhead surveying of the pit entrance, entering the pit, and determining the extent of the cave/lava tube. During an exploration mission, it may be strategic to place multiple robots along the cave/lava tube to maintain a direct line-of-sight connection link, which can be used for communication and wireless power transfer. The mission would continue even if one or more of the SphereX drones were lost or destroyed. The SphereX drone utilizes the rover or lander services such as the power to recharge and a long-distance communication link to Earth. As we have shown, this mother-daughter architecture compartmentalizes risk to the SphereX robots while benefiting from the rapid infusion of new technology and the ability to pursue high-risk, high reward science without compromising the overall mission. This proposed architecture opens whole new avenues for exploring off-world environments in the solar system and can even lay the foundation for a future high-risk and more sophisticated exploration mission.

## 9. Patents

Himangshu Kalita and Jekan Thangavelautham report a pending patent on Spherical Robots for Off-World Surface Exploration.

## Figures and Tables

**Figure 1 sensors-21-06203-f001:**
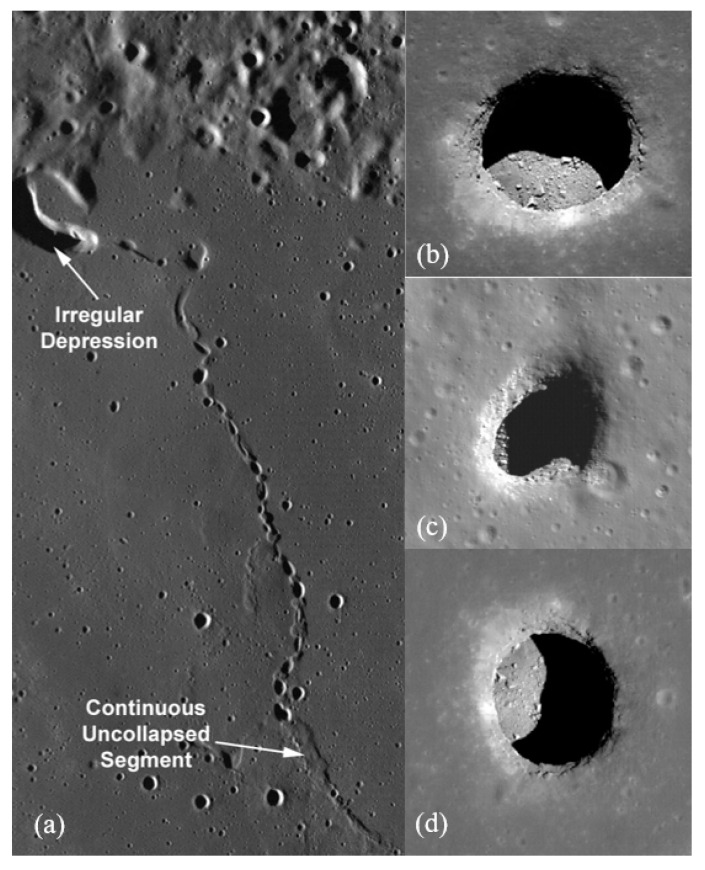
(**a**) Section showing the area where the feature transitions from a chain of collapse pits to a continuous unclasped segment (lava tube); (**b**) Marius Hill Hole (MHH); (**c**) Mare Tranquilitatis Hole (MTH); (**d**) Mare Ingenii Hole (MIH) [NASA/GSFC/Arizona State University].

**Figure 2 sensors-21-06203-f002:**
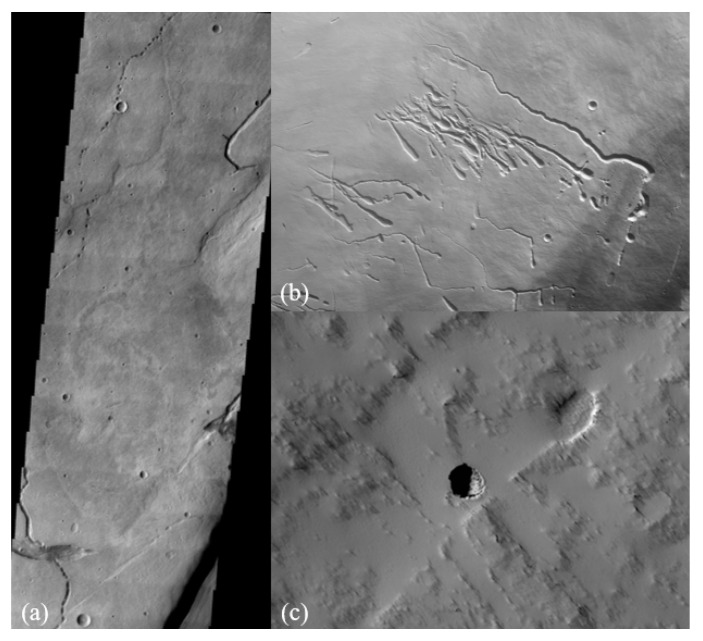
(**a**) THEMIS image of lava tube collapsed pits from Hadriaca Patera [NASA/JPL/Arizona State University]; (**b**) HRSC image of circular depressions, or pit chains along with linear channel features (lava tubes) on Pavonis Mons [ESA/DLR/FU Berlin]; (**c**) HiRISE image of pit entrance on Tartarus Colles [NASA/JPL/University of Arizona].

**Figure 3 sensors-21-06203-f003:**
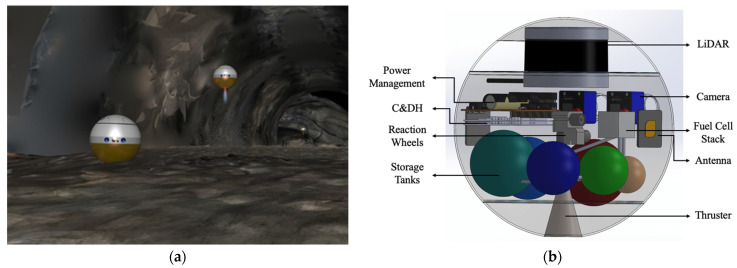
(**a**) Artistic view of SphereX robots cooperatively exploring underground pit on the Moon; (**b**) Preliminary 3D CAD model of SphereX.

**Figure 4 sensors-21-06203-f004:**
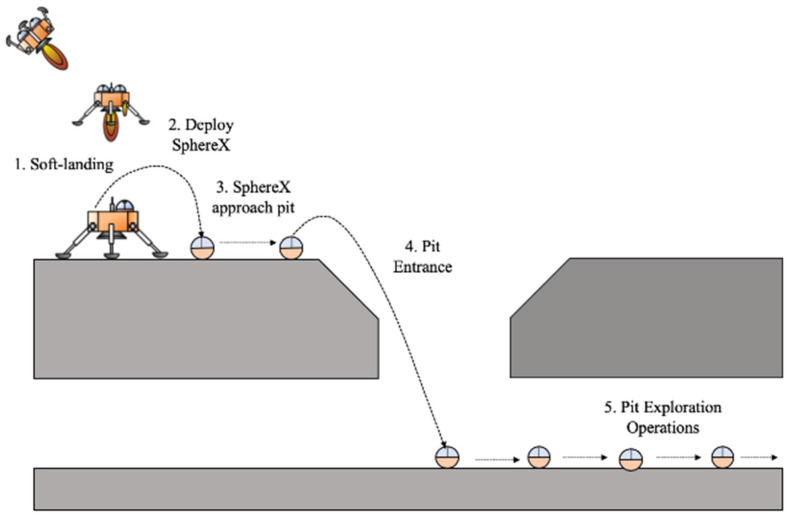
An example of a Lunar lander carrying multiple SphereX robots for pit exploration.

**Figure 5 sensors-21-06203-f005:**
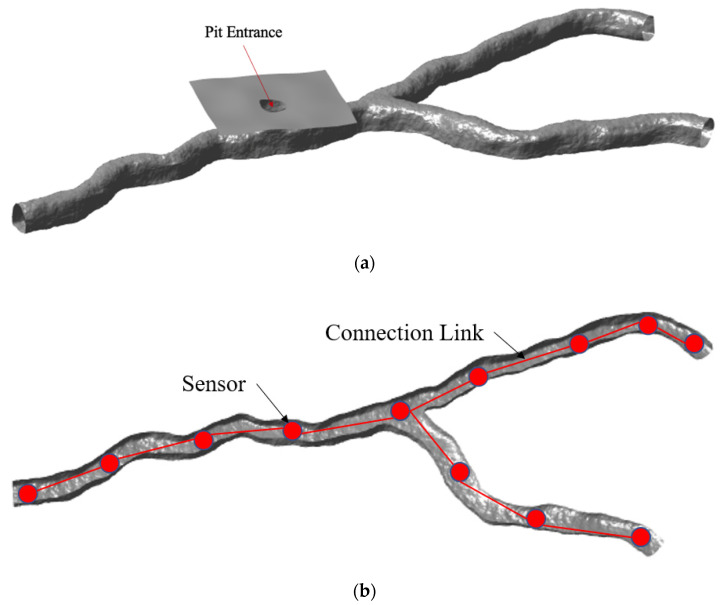
(**a**) 3D model of a cave with vertical entrance for analysis modeled in Blender software; (**b**) Placement of multiple sensors along the cave to maintain a direct line-of-sight connection link.

**Figure 6 sensors-21-06203-f006:**
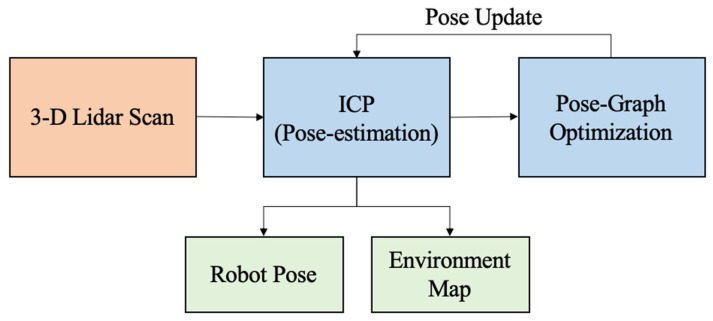
Framework for ICP-based Pose-graph SLAM algorithm for simultaneous localization and mapping.

**Figure 7 sensors-21-06203-f007:**
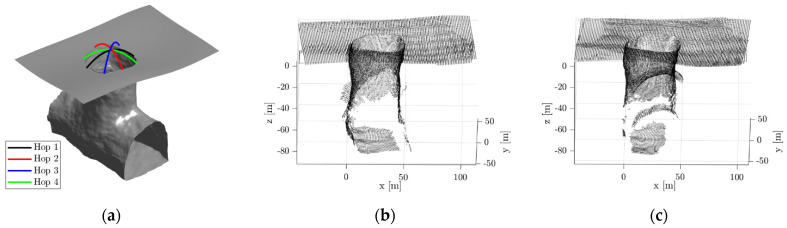
(**a**) Model of the pit entrance along with the trajectories of 4 hops used for mapping; (**b**–**e**) Local maps generated during Hop 1, Hop2, Hop3, and Hop 4; (**f**) Final map of the pit entrance.

**Figure 8 sensors-21-06203-f008:**
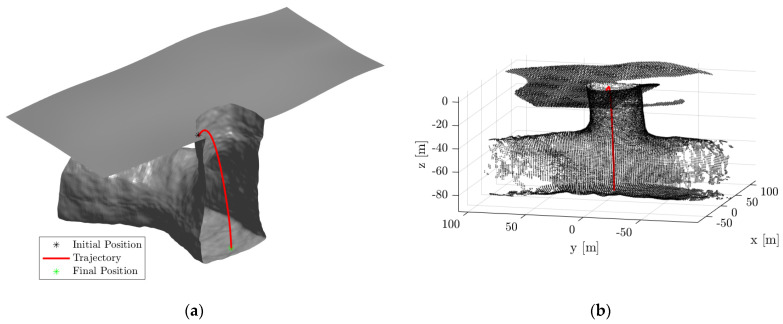
(**a**) Cut section of the pit entrance model showing the trajectory of SphereX to land inside the pit; (**b**) Map generated during the pit entrance trajectory along with the estimated trajectory of the robot shown in red; (**c**) True and estimated trajectory of the robot along x, y, and z-direction; (**d**) True and estimated quaternions of the robot.

**Figure 9 sensors-21-06203-f009:**
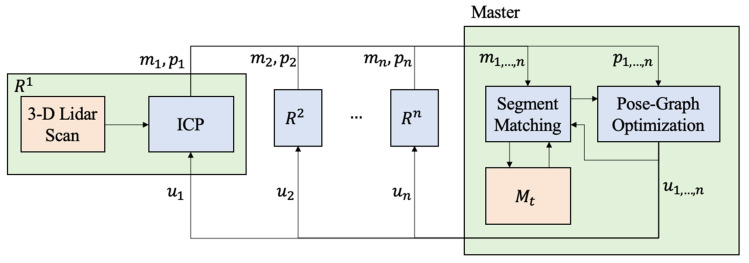
Framework for multi-robot mapping used to generate a master map $M_{t}$ of the cave, using local maps generated by each robot.

**Figure 10 sensors-21-06203-f010:**
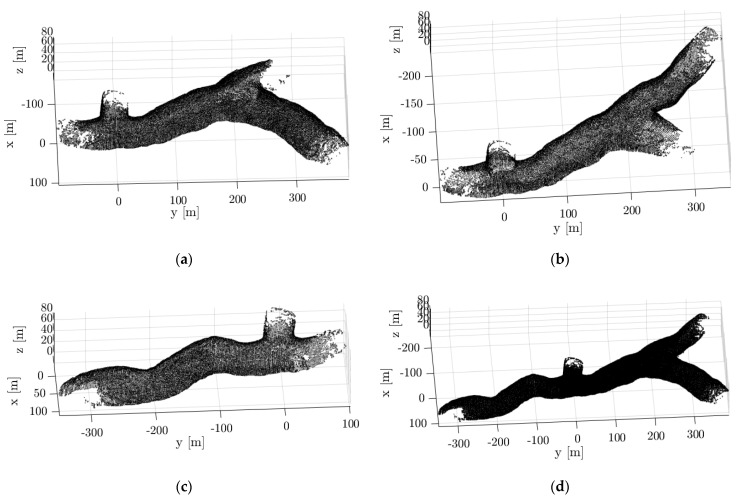
Map generated using the multi-robot mapping framework with 3 robots. (**a**–**c**) Map generated by robot 1, 2 and 3; (**d**) Master map.

**Figure 11 sensors-21-06203-f011:**
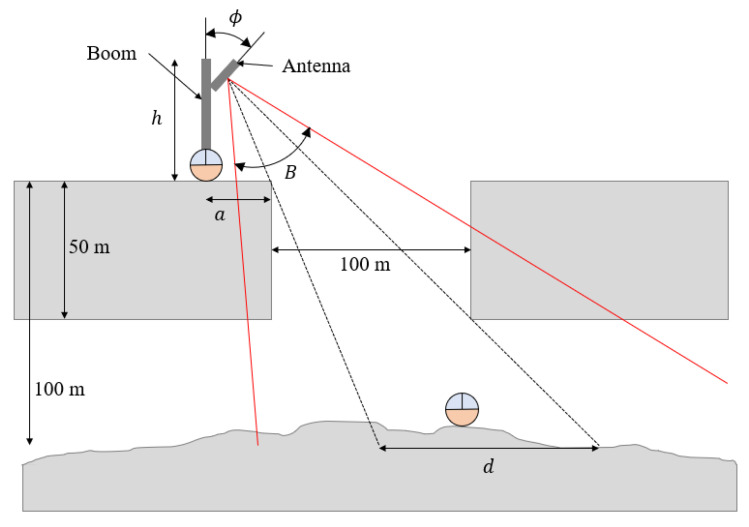
Schematic of the boom deployed by the SphereX on the surface near a pit entrance with an antenna at its tip. The dimensions are not scaled.

**Figure 12 sensors-21-06203-f012:**
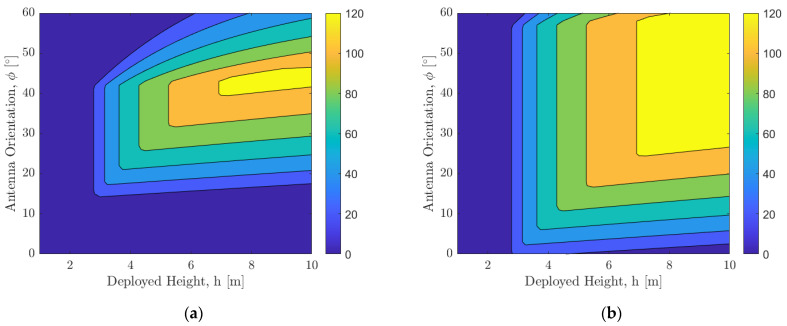
Variation of distance d in meters on the floor of the pit within which a direct line-of-sight communication link is possible with respect to the deployed boom height *h* and orientation of the antenna ϕ for (**a**) Beam width *B* = 30° and *a* = 5 m; (**b**) Beam width *B* = 60° and *a* = 5 m; (**c**) Beam width *B* = 30° and *a* = 10 m; (**d**) Beam width *B* = 60° and *a* = 10 m.

**Figure 13 sensors-21-06203-f013:**
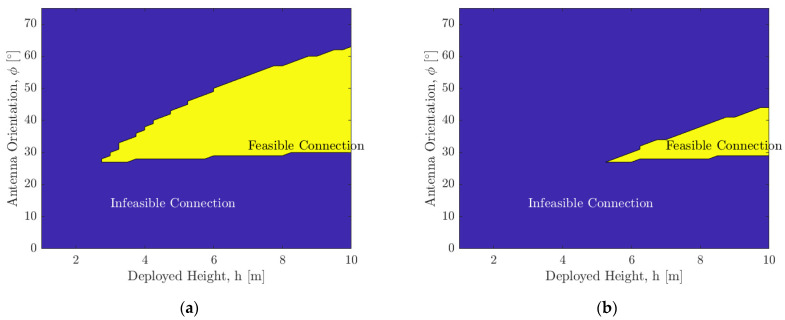
Variation of a feasible direct line-of-sight communication link for an optical transmitter with respect to the deployed boom height h in meters and orientation of the antenna ϕ in degrees for (**a**) a=5 m; (**b**) a=10 m.

**Figure 14 sensors-21-06203-f014:**
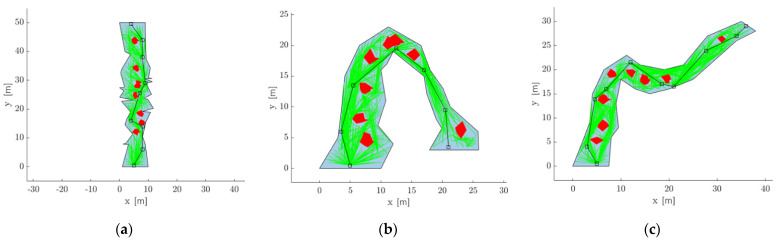
Placement of sensors for maintaining a direct line of sight link inside a cave represented by a non-convex polygon. The red polygons are the obstacles; the black squares are the sensors, and the lines connecting them are the direct line of sight connection link. (**a**) Cave model 1, (**b**) Cave model 2, and (**c**) Cave model 3.

**Figure 15 sensors-21-06203-f015:**
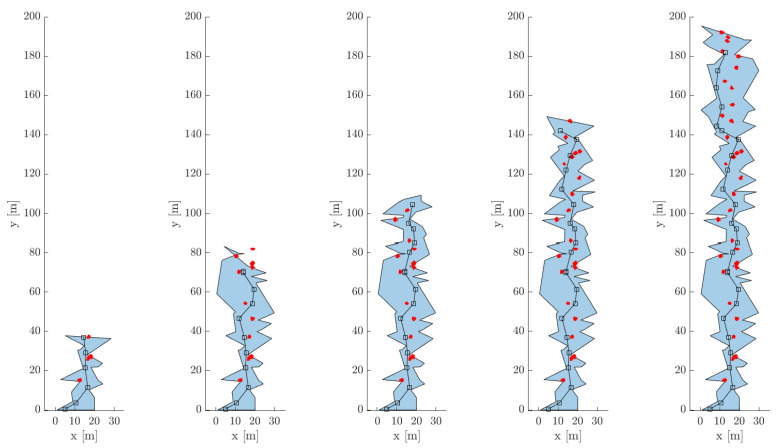
Placement of sensors inside an unknown cave over 5 successive Explore→Place Sensor cycles.

**Figure 16 sensors-21-06203-f016:**
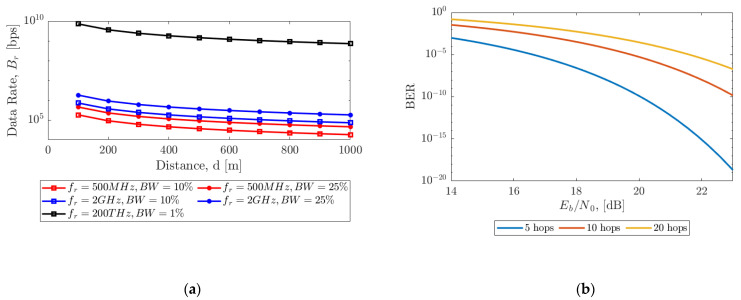
(**a**) Variation of data rate $B_{r}$ in bps over distance for 10 hops (each sensor placed at a distance of 100 m). The simulation is performed with RF communication (500 MHz and 2 GHz) with (10% and 25%) bandwidth and optical communication (200 THz) with (1%) bandwidth, (**b**) Variation of BER over $E_{b}/N_{0}$ for different number of hops.

**Figure 17 sensors-21-06203-f017:**
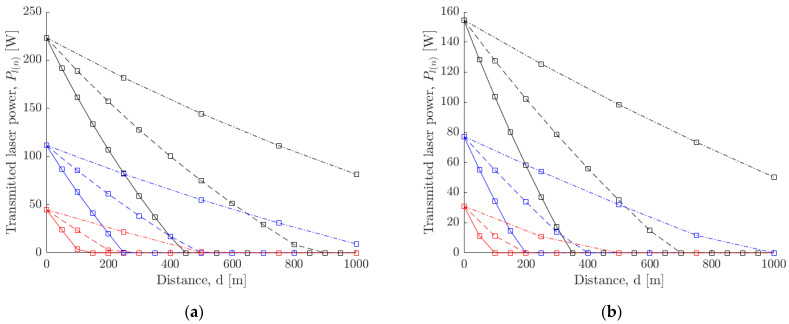
Variation of transmitted power by each robot placed at a distance d=[50, 100, 250] m, over a total distance of 1000 m for an initial supply power of $P_{s}=\left[100,\;250,\;500\right]$ W. The wavelength of the laser is (**a**) 810 nm, (**b**) 1550 nm.

**Table 1 sensors-21-06203-t001:** Legend for Figure 17.

LineSpec	Property
Color—‘red’	$P_{s}=100$ W
Color—‘blue’	$P_{s}=250$ W
Color—‘black’	$P_{s}=500$ W
LineStyle—‘Solid’	d=50 m
LineStyle—‘Dashed’	d=100 m
LineStyle—‘Dash-dot’	d=250 m

## Data Availability

Available upon request.

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
