# Peer review of "Strategies for Deploying a Sensor Network to Explore Planetary Lava Tubes"

_sensors, 2021, doi:10.3390/s21186203_

Round 1
Reviewer 1 Report
A few spelling mistakes, including in the first line "underlyinh".
The introduction is well written and interesting, would be nice to have some subsections headings in it, just to aid reading. In particular, where you move from motivation to the proposed robot.
It would be of interest to compare and contract this architecture against others briefly, for example a small tethered/winched daughter rover.
Where does the simulated lava tube come from? Based off an earth lava tube? Would be good to understand the likelihood of this type of structure.
Reviewer 2 Report
1.The main question addressed by the research is how multiple small, low cost robots that utilizing unconventional mobility can work as a team to explore environments that are unable to be explored by current Lunar and Marsian landers and rovers.
2.The topic is relevant in the field, because the increasing number of missions to the planets of the solar system creates such a demand, and so far there are no effective solutions.
3.Presented approach has elements of novelty and prospects for application
4.Specific improvements could the authors consider regarding the methodology are not necessary.
5.The conclusions are consistent with the evidence and arguments presented and they address the main question posed.
6.The references are appropriate.
7.Well done, good quality paper. However, some minor improvements are necessary:
(1)Description of the vertical axis in Fig. 7 should be added.
(2)Axle description in Fig. 10 should be added.
(3)In the descriptions of all axes on all graphs physical units should be given in square brackets instead of in round.
(4)The caption for Fig. 11 is misleading and should be changed. It cannot be agreed that this drawing is a description.
